# DEEP INVERSE REINFORCEMENT LEARNING VIA ADVERSARIAL ONE-CLASS CLASSIFICATION

## ABSTRACT

Traditional inverse reinforcement learning (IRL) methods require a loop to find the optimal policy for each reward update (called an inner loop), resulting in very time-consuming reward estimation. In contrast, classification-based IRL methods, which have been studied recently, do not require an inner loop and estimate rewards quickly, although it is difficult to prepare an appropriate baseline corresponding to the expert trajectory. In this study, we introduced adversarial one-class classification into the classification-based IRL framework, and consequently developed a novel IRL method that requires only expert trajectories. We experimentally verified that the developed method can achieve the same performance as existing methods.

## 1 INTRODUCTION

Inverse reinforcement learning (IRL) (Russell, 1998) refers to the problem of estimating rewards for reinforcement learning (RL) agents to acquire policies that can reproduce expert behavior. An RL algorithm learns a policy that maximizes the cumulative discounted reward under a given reward function. An IRL algorithm does the opposite; it estimates the reward from the given policies or trajectories to satisfy the condition under the assumption that the expert is maximizing the reward.

IRL has been applied in two main areas (Ramachandran & Amir, 2007). The first is apprenticeship learning, which enables the learning of complex policies for which it is difficult to design a reward function. Compared to behavioral cloning, IRL is robust to the covariate shift problem (Ross et al., 2011) and achieves superior performance even when the amount of data is small. The second is reward learning, where IRL is used to estimate rewards from the trajectory data of human and animal action sequences and to analyze the intention of the subject. In previous studies, IRL methods have been used to analyze human walking paths (Kitani et al., 2012) and the behavior of nematodes (Yamaguchi et al., 2018).

In traditional IRL methods, the IRL loop has an inner loop that computes the optimal policy for the reward being estimated until convergence. This inner loop presents a difficulty in applying IRL to tasks with a large state-action space because it is computation-intensive. As a solution to this, classification-based IRL methods transform the IRL problem into a problem of classifying the expert's trajectory and the trajectory to be compared. Notable methods include AIRL (Fu et al., 2017), LogReg-IRL (Uchibe, 2018), and T-REX (Brown et al., 2019).

These methods differ in the ways they are formulated, but they result in similar learning methods. Online methods, such as AIRL, collect the trajectories to be compared from the environment. Contrastingly, offline methods, such as LogReg-IRL and T-REX, collect the trajectories to be compared in advance, which enables them to further speed up and stabilize learning by not requiring access to the environment during training. However, the learning performance of current offline methods depends heavily on the properties of the trajectories to be compared or the ranking of the trajectories, which is difficult to collect.

In this study, we exploited the fact that the learning process of LogReg-IRL by binary classification is equivalent to that of a discriminator in adversarial learning, such as with generative adversarial networks (GANs) (Goodfellow et al., 2014). Specifically, we developed an innovative deep IRL method, called **state-only learned one-class classifier for IRL** (SOLO-IRL), in which binary classification is replaced with adversarial one-class classification. Figure 1 compares the traditional

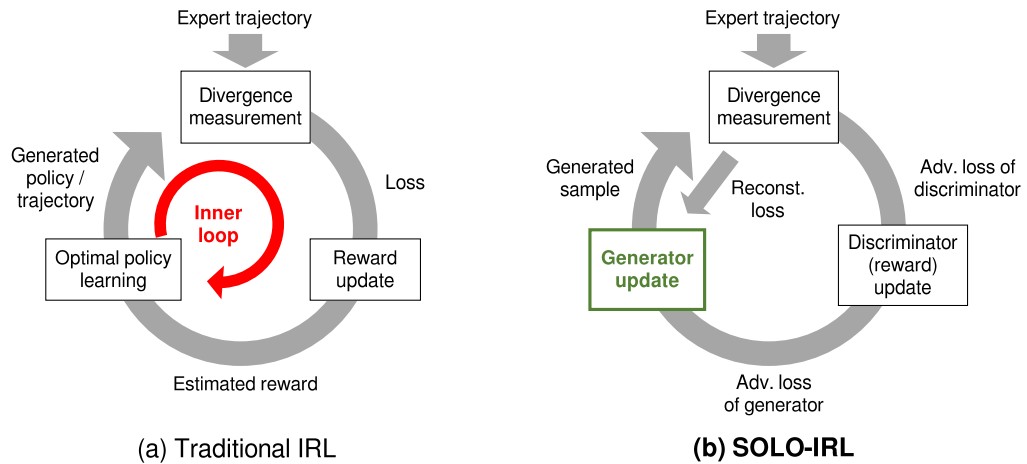

Figure 1: Comparison of traditional IRL and the proposed SOLO-IRL.

and proposed IRL methods. The proposed method does not require an inner loop and is an offline method; thus, it can be trained extremely fast. In addition, it does not require that trajectories be compared. With these advantages, the proposed method greatly advances the application of IRL methods to real-world problems.

## 2 PRELIMINARIES

### 2.1 MARKOV DECISION PROCESS (MDP)

RL is a learning problem based on the Markov decision process (MDP). The MDP consists of a tuple $M = \langle S, A, P, R, \gamma \rangle$, where $S$ is the state space, $A$ is the action space, $P$ is the state-transition probability, $R$ is the reward function, and $\gamma$ is the discount factor indicating the degree of importance for future rewards. In the MDP, the state-value function for state $s_t$ at time $t$ is represented by the Bellman equation, as follows:

$$V(s_t) = \max_a \left\{ R(s_t, a) + \sum_{s'} p(s'|s_t, a)\gamma V(s') \right\} \tag{1}$$

where $R(s_t, a_t)$ is the reward for taking action $a_t$ in state $s_t$ and $p(s_{t+1}|s_t, a_t)$ is the probability of transitioning to the next state $s_{t+1}$ when taking action $a_t$ in state $s_t$.

### 2.2 LINEARLY SOLVABLE MDP (LMDP)

The linearly solvable MDP (LMDP) is an extension of the MDP in which the agent directly determines the transition probability $u(s_{t+1}|s_t)$ from the current state $s_t$ to the next state $s_{t+1}$ as the control, instead of the action $a_t$ in the MDP. Then, the Bellman equation is linearized under two assumptions. First, the state-transition probability $p(s_{t+1}|s_t, u)$ is assumed to be expressed as the product of the uncontrolled transition probability $\bar{p}(s_{t+1}|s_t)$ and $u$ as follows:

$$p(s_{t+1}|s_t, u(s_{t+1}|s_t)) = \bar{p}(s_{t+1}|s_t) \exp \{u(s_{t+1}|s_t)\} \tag{2}$$

The uncontrolled transition probability $\bar{p}(s_{t+1}|s_t)$ indicates a transitional relationship between the states in the environment. When a transition is impossible, i.e., $\bar{p} = 0$, then $p = 0$.

The second assumption is that the reward $R(s_t, u)$ is composed of a state-dependent reward $r(s_t)$ and penalty term $D_{\mathrm{KL}}(p||\bar{p})$ for state-transition probability $p$ over the divergence from the uncontrolled transition probability $\bar{p}$. This assumption can be formulated as follows:

$$R(s_t, u(s_{t+1}|s_t)) = r(s_t) - D_{\mathrm{KL}}(p(s_{t+1}|s_t, u(s_{t+1}|s_t))||\bar{p}(s_{t+1}|s_t)) \tag{3}$$

where $D_{\mathrm{KL}}(P_x||P_y)$ represents the Kullback–Leibler (KL) divergence of $P_x$ and $P_y$. By rearranging Eq. (3) according to the definition of the KL divergence, the following equation is obtained:

$$R\left(s_t, u(s_{t+1}|s_t)\right) = r(s_t) - \sum_{s'} p(s'|s_t, u(s'|s_t))u(s'|s_t) \tag{4}$$

Substituting Eq. (4) into the Bellman equation in Eq. (1) gives the following:

$$V(s_t) = r(s_t) + \max_u \left\{ \sum_{s'} p(s'|s_t, u(s'|s_t)) \Big[ -u(s'|s_t) + \gamma V(s') \Big] \right\} \tag{5}$$

Eq. (2) is then substituted into Eq. (5) and the Lagrange multiplier applied with $\sum_{s'} p(s'|s_t, u) = 1$ as a constraint. Finally, the max operator is removed, resulting in the linear Bellman equation as follows:

$$\exp\{V(s_t)\} = \exp\{r(s_t)\} \sum_{s'} \bar{p}(s'|s_t) \exp\{\gamma V(s')\} \tag{6}$$

The optimal control $u^*$ in the LMDP is given by

$$u^*(s_{t+1}|s_t) = \frac{\bar{p}(s_{t+1}|s_t) \exp\{\gamma V(s_{t+1})\}}{\sum_{s'} \bar{p}(s'|s_t) \exp\{\gamma V(s')\}} \tag{7}$$

### 2.3 Logistic Regression-Based IRL (LogReg-IRL)

LogReg-IRL (Uchibe, 2018) is a deep IRL method in the LMDP. The following is an overview of the IRL framework in LogReg-IRL. By rearranging the linear Bellman equation in Eq. (6), the following is obtained:

$$\exp\{V(s_t) - r(s_t)\} = \sum_{s'} \bar{p}(s'|s_t) \exp\{\gamma V(s')\} \tag{8}$$

Then, substituting Eq. (8) into Eq. (7) and rearranging the result gives

$$
\begin{aligned}
u^*(s_{t+1}|s_t) &= \frac{\bar{p}(s_{t+1}|s_t) \exp\{\gamma V(s_{t+1})\}}{\exp\{V(s_t) - r(s_t)\}} \\
\frac{u^*(s_{t+1}|s_t)}{\bar{p}(s_{t+1}|s_t)} &= \exp\{r(s_t) + \gamma V(s_{t+1}) - V(s_t)\} \\
\log \frac{u^*(s_{t+1}|s_t)}{\bar{p}(s_{t+1}|s_t)} &= r(s_t) + \gamma V(s_{t+1}) - V(s_t)
\end{aligned}
\tag{9}
$$

Applying Bayes' theorem to Eq. (9) we obtain

$$\log \frac{u^*(s_t, s_{t+1})}{\bar{p}(s_t, s_{t+1})} = \log \frac{u^*(s_t)}{\bar{p}(s_t)} + r(s_t) + \gamma V(s_{t+1}) - V(s_t) \tag{10}$$

The left-hand side and the first term on the right-hand side of Eq. (10) are the density-ratios. The density-ratio $p_a/p_b$ can be estimated by assigning the label $\eta = 1$ to the samples from the probability distribution $p_a$, assigning $\eta = -1$ to the samples from $p_b$, and training a classifier using logistic regression (Qin, 1998; Cheng et al., 2004; Bickel et al., 2007). First, by Bayes' theorem, the following is obtained:

$$
\begin{aligned}
\frac{p_a(x)}{p_b(x)} &= \frac{p(\eta = 1|x)}{p(\eta = -1|x)} \frac{p(\eta = -1)}{p(\eta = 1)} \\
\log \frac{p_a(x)}{p_b(x)} &= \log \frac{p(\eta = 1|x)}{p(\eta = -1|x)} + \log \frac{p(\eta = -1)}{p(\eta = 1)}
\end{aligned}
\tag{11}
$$

Next, the first discriminator $D_1(x)$ is defined by the sigmoid function $\sigma(x) = 1/\{1 + \exp(-x)\}$ and a neural network $f(x)$:

$$D_1(x) = p(\eta = 1|x) = \sigma(f(x)) \tag{12}$$

where the second term on the right-hand side of Eq. (11) can be approximated by calculating the sample number ratio $N_{p_a}/N_{p_b}$ and taking its logarithm. For the first term, the following equation can be obtained from the definition of the discriminator in Eq. (12):

$$
\begin{aligned}
\log \frac{p(\eta = 1|x)}{p(\eta = -1|x)} &= \log \frac{D_1(x)}{1 - D_1(x)} \\
&= \log \frac{1 + \exp\{f(x)\}}{1 + \exp\{-f(x)\}} \\
&= \log \exp\{f(x)\} \\
&= f(x)
\end{aligned}
\tag{13}
$$

From Eq. (13), when $N_{p_a} = N_{p_b}$, the following holds:

$$
\log \frac{p_a(x)}{p_b(x)} = f(x)
\tag{14}
$$

Therefore, the density-ratio of the first term in Eq. (10) can be estimated by sampling the states $s_t^* \sim \tau^*$ and $\bar{s}_t \sim \bar{\tau}$ from the expert trajectory $\tau^*$ according to the optimal control $u^*$ and the baseline trajectory $\bar{\tau}$ according to the uncontrolled transition probability $\bar{p}$, followed by training with the following cross-entropy loss:

$$
L_1(D_1) = -\mathbb{E}_{\bar{s}_t \sim \bar{\tau}}[\log(1 - D_1(\bar{s}_t))] - \mathbb{E}_{s_t^* \sim \tau^*}[\log(D_1(s_t^*))]
\tag{15}
$$

The density-ratio on the left-hand side of Eq. (10) is defined as follows using the trained $f(x)$, reward-estimating neural network $\tilde{r}(x)$, and state-value-estimating neural network $\tilde{V}(x)$:

$$
\log \frac{u^*(s_t, s_{t+1})}{\bar{p}(s_t, s_{t+1})} = f(s_t) + \tilde{r}(s_t) + \gamma \tilde{V}(s_{t+1}) - \tilde{V}(s_t)
\tag{16}
$$

The second discriminator $D_2$ for the state-transition pair is defined as

$$
D_2(x, y) = \sigma(f(x) + \tilde{r}(x) + \gamma \tilde{V}(y) - \tilde{V}(x))
\tag{17}
$$

As with $D_1$, the discriminator $D_2$ is trained by cross-entropy loss $L_2$, given as

$$
L_2(D_2) = -\mathbb{E}_{(\bar{s}_t, \bar{s}_{t+1}) \sim \bar{\tau}}[\log(1 - D_2(\bar{s}_t, \bar{s}_{t+1}))] - \mathbb{E}_{(s_t^*, s_{t+1}^*) \sim \tau^*}[\log(D_2(s_t^*, s_{t+1}^*))]
\tag{18}
$$

In the original LogReg-IRL, an L2 regularization term is added to the loss function. Following the process described above, LogReg-IRL estimates the reward and state-value by classifying the expert and baseline trajectories. Unlike traditional IRL methods, LogReg-IRL does not require RL in the reward estimation process and, thus, it can be trained very quickly.

## 2.4 DIFFICULTY COLLECTING BASELINE TRAJECTORIES

LogReg-IRL showed that IRL in LMDP can be formulated by learning two discriminators. However, LogReg-IRL has a problem in that its learning performance is greatly affected by the baseline trajectory. For the baseline trajectory, it is desirable to collect data that follow uncontrolled transition probability $\bar{p}$, such as trajectories obtained under a random policy, for a wide range of states.

However, for some tasks, the number of states that can be transitioned by a random policy may be limited. For example, in a game task, such as an Atari game, the game does not progress, and in a driving simulator task such as TORCS (Wymann et al., 2000), the car crashes into a wall immediately. In such cases, data according to a random policy cannot cover a wide range of states.

Several methods have been proposed to collect the baseline trajectory in LogReg-IRL. For Atari games, a method using state-transition pairs from random policy in any state in the expert trajectory was proposed (Uchibe, 2018). For TORCS, a method using the trajectory recorded by driving with noise added to the action output of the expert agent was proposed (Kishikawa & Arai, 2021). However, those proposed methods are task-specific, and there is no well-established generalized method for collecting baseline trajectories.

An inappropriate baseline trajectory leads to inappropriate reward estimation. The density-ratios diverge with respect to the states that the agent can reach and those for which there is no baseline trajectory, and high rewards are estimated where there are no experts. Therefore, an agent that learns according to the estimated reward may acquire a different action from the expert as the optimal policy.

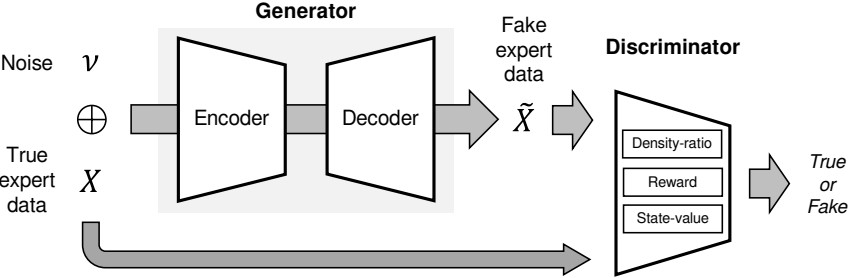

Figure 2: Proposed SOLO-IRL.

# 3  STATE-ONLY LEARNED ONE-CLASS CLASSIFIER FOR IRL (SOLO-IRL)

We propose the novel IRL method SOLO-IRL, which estimates the reward given only the expert. SOLO-IRL is a combination of an IRL framework based on LMDP, a transition generator based on adversarial one-class classification, and least-squares loss. Each of these is explained below.

## 3.1  SOLUTION BY ADVERSARIAL ONE-CLASS CLASSIFICATION

Classification-based IRL methods are equivalent to learning a discriminator in adversarial learning frameworks, such as GANs (Goodfellow et al., 2014), which are binary classification problems. This means that the IRL problem can be solved by learning a discriminator that classifies the trajectory as expert or not.

Broadening our perspective to other fields, we find that the anomaly detection problem also requires a binary classifier to distinguish between normal and abnormal samples. However, in real problems, we often encounter situations in which we can obtain many normal samples but few abnormal samples. Therefore, one-class classification is a method for obtaining a binary classifier using only normal samples.

Recently, the adversarially learned one-class classifier (ALOCC) (Sabokrou et al., 2018) was proposed as a one-class classification method. In the ALOCC, the discriminator is adversarially trained with a denoising autoencoder that generates fake normal samples. Consequently, the discriminator is trained as a binary classifier that identifies normal and abnormal samples using only normal samples.

SOLO-IRL combines a classification-based IRL method with the ALOCC. The structure of SOLO-IRL is illustrated in Figure 2, and its algorithm is given as Algorithm 1. In the following, we describe the learning process and features of SOLO-IRL as well as the objective function used for learning SOLO-IRL.

## 3.2  LEARNING PROCESS OF SOLO-IRL

The training of SOLO-IRL consists of two stages. In the first stage, we learn a discriminator $D_1$ that classifies whether a state is sampled from expert data or not. Generator $G_1$ is composed of an encoder–decoder network $\mathcal{R}_1$ and generates a fake current state $\bar{s}_t$ from the true current state $s_t^*$ plus noise $\nu_1$, as shown in Eq. (19). Then, the generator learns such that the generated $\bar{s}_t$ and $s_t^*$ are close, and $D_1$ judges $\bar{s}_t$ to be the true current state.

$$\bar{s}_t \leftarrow \mathcal{R}_1(s_t^* + \nu_1) \tag{19}$$

Meanwhile, discriminator $D_1$ is defined by Eq. (12) and outputs the probability that a given state is sampled from the expert. $D_1$ learns to distinguish between states sampled from a true expert and a fake expert from a generator. Here, the density-ratio representing the expertness of each state is obtained in $D_1$.

In the second stage, generator $G_2$ learns to generate a state-transition pair that is close to the sampled expert data and that discriminator $D_2$ judges as expert. Generator $G_2$ uses two encoder–decoder networks $\mathcal{R}_2$ and $\mathcal{R}_3$ to generate a fake state-transition pair $(\bar{s}_t, \bar{s}_{t+1})$ from the expert's current state $s_t^*$ plus noise $\nu_2$, as given by Eqs. (20) and (21). Then, the generator learns such that the generated $(\bar{s}_t, \bar{s}_{t+1})$ and $(s_t^*, s_{t+1}^*)$ are close to each other, and discriminator $D_2$ judges the generated data to be a true state-transition pair.

$$\bar{s}_t \quad \leftarrow \quad \mathcal{R}_2(s_t^* + \nu_2) \tag{20}$$

$$\bar{s}_{t+1} \quad \leftarrow \quad \mathcal{R}_3(s_t^* + \nu_2) \tag{21}$$

The training of $D_2$ is the same as that of $D_1$ in the first stage. Here, discriminator $D_2$ contains a reward network $\tilde{r}$ and state-value network $\tilde{V}$, as shown in Eq. (17). Finally, it works as an IRL algorithm to estimate the reward and state-value.

By introducing a generator and training the discriminator in an adversarial manner, the decision boundary around the expert is refined, and the appropriate reward is estimated. This makes preparation of the baseline trajectory by trial and error unnecessary. In addition, because SOLO-IRL is an offline method, it learns quickly without executing RL or interacting with the environment. To the best of our knowledge, SOLO-IRL is the only method that estimates the reward and state-value exclusively from expert trajectories.

### 3.3 LEAST-SQUARES LOSS AS ADVERSARIAL OBJECTIVE

To train the generator and discriminator, we propose using least-squares loss as an adversarial objective. The least-squares loss is represented by the following equations:

$$L_{adv}(D) \quad = \quad \frac{1}{2}\mathbb{E}_{x \sim d^*}[(D(x) - 1)^2] + \frac{1}{2}\mathbb{E}_{\tilde{x} \sim \bar{d}}[(D(\tilde{x}))^2] \tag{22}$$

$$L_{adv}(G) \quad = \quad \frac{1}{2}\mathbb{E}_{\tilde{x} \sim \bar{d}}[(D(\tilde{x}) - 1)^2] \tag{23}$$

where $d^*$ and $\bar{d}$ denote the true and fake states or state-transition pairs, respectively. The least-squares loss was proposed alongside LSGAN (Mao et al., 2017), which solved the learning instability and mode collapse problems of previous GANs. In SOLO-IRL, this least-squares loss is used instead of the cross-entropy loss to address the problems of learning stability and mode collapse.

### 3.4 RECONSTRUCTION OBJECTIVE

Meanwhile, the generators $G_1$ and $G_2$ must be trained to be close to the true expert data in addition to training by the adversarial objective. The reconstruction objective is trained by the least-squares loss given by the following equations, as in the original ALOCC:

$$L_{rec}(G_1) \quad = \quad ||s_t^* - \mathcal{R}_1(s_t^* + \nu_1)||_2 \tag{24}$$

$$L_{rec}(G_2) \quad = \quad ||s_t^* - \mathcal{R}_2(s_t^* + \nu_2)||_2 + ||s_{t+1}^* - \mathcal{R}_3(s_t^* + \nu_2)||_2 \tag{25}$$

Finally, the objective of discriminators $D_1$ and $D_2$ becomes $L_{adv}(D)$, and the objective of generators $G_1$ and $G_2$ becomes $L_{adv}(G) + L_{rec}(G)$. Using these objectives, SOLO-IRL estimates the reward and state-value by training each neural network using the gradient descent method.

## 4 EXPERIMENTAL RESULTS AND DISCUSSION

We validated the performance of the proposed method in the OpenAI Gym environment (Brockman et al., 2016). For the expert trajectory, we used a trajectory generated by an agent that had learned the optimal action for the true reward as an expert. As RL algorithms, we used PPO (Schulman et al., 2017) for the tasks with a discrete action space and TD3 (Fujimoto et al., 2018) for the tasks with a continuous action space.

In SOLO-IRL, a three-layer multilayer perceptron was used as both the encoder and decoder in the generator, and a three-layer multilayer perceptron was used as the discriminator. Adam (Kingma & Ba, 2014) was used to optimize the neural network.

---

**Algorithm 1** SOLO-IRL: State-Only Learned One-class Classifier for IRL

---

**Require:** Expert trajectories $\tau^*$, discount factor $\gamma$, noise $\nu_1$ and $\nu_2$, numbers of iterations $n_1$ and $n_2$
**Ensure:** Reward network $\tilde{r}(x)$, state-value network $\tilde{V}(x)$
 1: Initialize neural network $f(x)$, $\mathcal{R}_1(x)$
 2: Define discriminator $D_1(s_t) = \sigma(f(s_t))$
 3: **for** $i = 0, \cdots, n_1$ **do**
 4:     Sample expert state $s_t^*$ from $\tau^*$
 5:     $\bar{s}_t \leftarrow \mathcal{R}_1(s_t^* + \nu_1)$
 6:     Train $D_1$ according to the loss $L_{adv}(D)$ in Eq. (22)
 7:     Train $\mathcal{R}_1$ with loss $L_{adv}(G) + L_{rec}(G_1)$ in Eqs. (23) and (24)
 8: **end for**
 9: Initialize neural network $\tilde{r}(x)$, $\tilde{V}(x)$, $\mathcal{R}_2(x)$, $\mathcal{R}_3(x)$
10: Define discriminator $D_2(s_t, s_{t+1}) = \sigma(f(s_t) + \tilde{r}(s_t) + \gamma\tilde{V}(s_{t+1}) - \tilde{V}(s_t))$
11: Disable $f$ updates
12: **for** $i = 0, \cdots, n_2$ **do**
13:     Sample expert current state $s_t^*$ and next state $s_{t+1}^*$ from $\tau^*$
14:     $\bar{s}_t \leftarrow \mathcal{R}_2(s_t^* + \nu_2)$
15:     $\bar{s}_{t+1} \leftarrow \mathcal{R}_3(s_t^* + \nu_2)$
16:     Train $D_2$ according to the loss $L_{adv}(D)$ in Eq. (22)
17:     Train $\mathcal{R}_2$ and $\mathcal{R}_3$ according to the loss $L_{adv}(G) + L_{rec}(G_2)$ in Eqs. (23) and (25)
18: **end for**

---

Table 1: Validation results for the OpenAI Gym tasks. These scores are cumulative true rewards (averaged over 1000 trials); the higher the better.

|  | CartPole | BipedalWalker | Hopper | Walker2d |
|---|---|---|---|---|
| Expert | 499.96 | 321.69 | 3682.28 | 5272.42 |
| Random | 22.50 | −99.61 | 17.87 | 1.87 |
| LogReg-IRL (Uchibe, 2018) | 498.42 | −118.85 | 5.30 | 376.44 |
| SOLO-IRL (**ours**) | 449.11 | 226.16 | 1084.46 | 774.87 |
| Behavioral cloning | 500.00 | 135.13 | 3677.99 | 4920.64 |
| RED (Wang et al., 2019) |  |  | 3633.72 | 3868.98 |

We compared SOLO-IRL to LogReg-IRL (Uchibe, 2018). The baseline trajectory in LogReg-IRL was collected by running a random policy based on a uniform distribution in the environment. The LogReg-IRL implementation was created based on the SOLO-IRL implementation with the following changes: removal of the generator, changing of the cross-entropy loss, and addition of a weight decay term (coefficient: 1e-3) to the loss.

We used the following four tasks for validation.

**CartPole.** CartPole (Gym, b) is a basic RL task in which the agent must keep a pendulum connected to a cart upright by controlling the cart. A survival reward is given to the pendulum as long as it remains upright.

**BipedalWalker.** BipedalWalker (Gym, a) is a task in which a robot with four joints learns a bipedal walking task. The agent is required to output the velocities of its two hips and knees as actions, taking the state given by the virtual sensor as the input. The true reward comprises a bonus based on the speed and a penalty based on the magnitude of the action or fall.

**MuJoCo.** MuJoCo (Todorov et al., 2012) is an environment that collects tasks for the development of robotics. We selected Hopper and Walker2d as examples. Hopper is a task in which a one-legged robot moves forward, and Walker2d is one in which a two-legged robot walks. These two tasks require more complex continuous control than BipedalWalker.

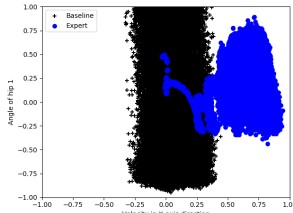 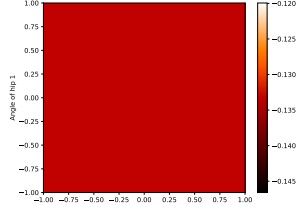 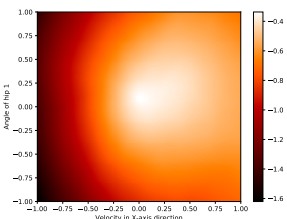

Figure 3: Visualization of the distribution of expert trajectories (blue) and baseline trajectories (black).

Figure 4: Visualization of the reward estimated by LogReg-IRL.

Figure 5: Visualization of the reward estimated by SOLO-IRL.

**Analysis of results.** First, in each environment, we evaluated the policy of the expert agent, the random policy, the policy of the agent trained according to the reward estimated by LogReg-IRL and SOLO-IRL, and the policy obtained via behavioral cloning. We also tested the random expert distillation (RED) (Wang et al., 2019), which is a recent method of imitation learning in the MuJoCo environment[1].

There are two metrics for evaluating the performance of IRL: expected value difference (EVD) (Levine et al., 2011) and expected cumulative reward. To calculate the EVD, we used the discount rate. However, since the appropriate discount rate varies depending on the environment, the value we use will also vary accordingly. Therefore, we decided to evaluate the performance using the expected cumulative reward, which is a more common of the two.

The results are presented in Table 1. The results for CartPole show that SOLO-IRL was able to achieve comparable with that of LogReg-IRL. Furthermore, with regard to the results for Bipedal-Walker, we saw that LogReg-IRL failed to learn, whereas SOLO-IRL obtained a good score.

The aforementioned results can be attributed to the fact that BipedalWalker has a vast state space, whereas CartPole has a relatively small state space. Consequently, the random policy could only collect data near the starting point and could not provide an appropriate baseline trajectory for the expert trajectory that progressed to the goal.

Behavioral cloning and RED generally perform better than the other methods that were used in the comparison. However, in BipedalWalker, the proposed method showed better performance than behavioral cloning due to the changing environment. In addition, RED requires more information than the proposed method because of the availability of RL. Therefore, our proposed method can be said to be superior than the traditional methods in that it can learn purely using expert trajectories.

**Relationship between trajectory distribution and estimated reward in the BipedalWalker task.** Among the 24 dimensions in the BipedalWalker state, Figure 3 visualizes the distribution of the expert and baseline trajectories for "velocity in the X-axis direction" and "angle of hip 1," and Figures 4 and 5 visualize the estimated reward for these two dimensions at the starting point. As can be seen, the baseline trajectory does not cover a part of the the expert trajectory. Owing to the lack of a baseline trajectory, LogReg-IRL's estimation failed, yielding a meaningless reward. In contrast, SOLO-IRL's estimation, which was trained via generating samples near the expert, resulted in an appropriate reward that drove the agent forward.

**Behavior of adversarial and reconstruction objectives.** Figures 6 and 7 illustrate the behavior of adversarial loss and reconstruction loss during training in SOLO-IRL. With adversarial learning, the probabilities of truthfulness of the true and fake samples converge to an equilibrium near 0.5. The discriminator narrows the decision boundary toward the expert neighborhood through learning, and the generator eventually succeeds in producing samples that are close to the true. The learning success of the generator is also evident from the convergence of the reconstruction loss.

**Summary of the experimental results.** There is room for improving the performance of LogReg-IRL using complex rule-based policies or by combining multiple policies to collect baseline tra-

---

[1]Since the author's implementation of RED only supported MuJoCo, the experiments were also conducted only on the MuJoCo task.

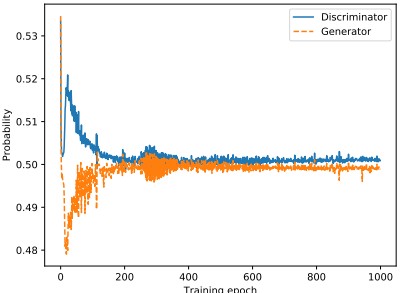

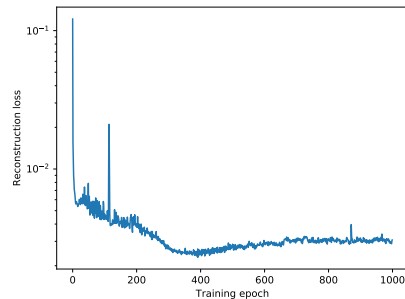

Figure 6: Changes in the probability that the true sample (Discriminator) and fake sample (Generator) are true, respectively, during training in the first stage.

Figure 7: Reconstruction loss during training in the first stage.

jectories; however, this is expected to be significantly more difficult than adjusting the noise in SOLO-IRL. Therefore, it can be said that similar or better performance than LogReg-IRL can be achieved more easily using SOLO-IRL.

## 5  RELATED WORKS

**IRL.** A method similar to the proposed method is the AIRL, which is an extension of maximum entropy IRL (Ziebart et al., 2008), guided cost learning (Finn et al., 2016), and generative adversarial imitation learning (GAIL) (Ho & Ermon, 2016). The theoretical background of the aforementioned methods is related to the proposed method in terms of entropy regularization (Chow et al., 2018). AIRL requires access to the environment, while the proposed method does not. In addition, D-REX (Brown et al., 2020) is similar to the proposed method in that it creates the trajectory of the comparison target by adding noise. Since the proposed method uses adversarial learning, it can generate more appropriate samples for comparison and does not need to learn behavioral cloning.

**Imitation learning.** There are several methods that have been recently proposed for imitating expert demonstrations, such as RED, disagreement-regularized imitation learning (Brantley et al., 2019), and O-NAIL (Arenz & Neumann, 2020). All these methods are extensions of GAIL and require information regarding the expert's state-action pairs. The proposed method can be used to estimate the reward from the trajectory comprising only states.

**Behavioral cloning.** The simplest offline learning method to imitate expert trajectory data is behavioral cloning, and (Torabi et al., 2018) is such a method that can be applied to state-only trajectories. However, compared to IRL methods, it is difficult to deal with the issues of policy transferability and covariate shift using such a method.

## 6  CONCLUSION

In this study, we exploited the fact that the classification-based IRL framework is equivalent to training a discriminator in adversarial learning and developed SOLO-IRL, in which a generator is incorporated to generate fake expert data. SOLO-IRL can be trained quickly without the need for an inner loop and easily estimates rewards with high performance exclusively from expert trajectories, with no need for baseline trajectories.

However, although it is simpler than collecting baseline trajectories, the noise used for reconstruction by the generator still requires adjustment. In future work, we will develop a method that does not require noise adjustment. We will also consider application to more advanced image-based control tasks and the incorporation of recent advances in GANs.

## REPRODUCIBILITY STATEMENT

In Appendices, we have described the details of the hyperparameters and noise needed to reproduce our experiments. We also attached the source code used in our experiments.

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

## A HYPERPARAMETERS OF RL

This section describes the main hyperparameters used in the experiments. For RL, we used the same settings for both the training of the expert agent and the training based on IRL results.

### A.1 HYPERPARAMETERS OF TD3 AGENT

The implementation of TD3 is based on `https://github.com/chainer/chainerrl/blob/master/examples/mujoco/reproduction/td3/train_td3.py`, which is identical except for the environment name and the values listed in Table 2.

Table 2: Some of the hyperparameters employed in training the TD3 agent.

| Hyperparameter | Value |
|---|---|
| Training steps | $10^6$ |
| Initial exploration sample size | $10^4$ |
| Replay buffer size | $10^6$ |
| Batch size | $10^3$ |
| Discount rate | 0.99 |

### A.2 HYPERPARAMETERS OF PPO AGENT

The implementation of PPO is based on `https://github.com/chainer/chainerrl/blob/master/examples/mujoco/train_ppo_gym.py`, which is identical except for the environment name and the values listed in Table 3.

Table 3: Some of the hyperparameters employed in training the PPO agent.

| Hyperparameter | Value |
|---|---|
| Training steps | $10^6$ |
| Update interval | 2048 |
| Batch size | 64 |
| Entropy coefficient | 0.001 |
| Discount rate | 0.99 |

## B IRL HYPERPARAMETERS

The hyperparameters employed in training SOLO-IRL and LogReg-IRL are shown in Table 4.

Table 4: Hyperparameters employed in training IRL.

| Hyperparameter | Value |
|---|---|
| Learning rate for network $f$ | 0.00004 |
| Learning rate for network $\tilde{r}$ | 0.00004 |
| Learning rate for network $\tilde{V}$ | 0.00004 |
| Learning rate for network $\mathcal{R}_1$ | 0.0001 |
| Learning rate for network $\mathcal{R}_2$ | 0.0001 |
| Learning rate for network $\mathcal{R}_3$ | 0.0001 |
| Number of inputs and outputs for each network layer | $(|S|, 1024), (1024, 1024),$ $(1024, 1024), (1024, 1)$ |
| Probability of dropout in each layer of discriminator | 0.0, 0.7, 0.7, 0.0 |
| Probability of dropout in each layer of generator | 0.0, 0.0, 0.0, 0.0 |
| Activation function in each layer | leaky ReLU, leaky ReLU, leaky ReLU, None |
| Number of training epochs in the first stage | 1000 |
| Number of training epochs in the second stage | 1000 |
| Number of steps in one epoch in the first stage | 100 |
| Number of steps in one epoch in the second stage | 100 |
| Batch size | 1024 |
| Discount rate | 0.99 |

## B.1 Additive noise

For CartPole, we added noise that follows a normal distribution $N(0, 0.001)$ to the true state. For the MuJoCo task, we added a normal distribution $N(0, 0.00001)$ noise to Hopper and a normal distribution $N(0, 0.1)$ noise to Walker2d.

For each dimension of BipedalWalker, we added either noise based on the normal distribution or noise for the labels. The tuning results for the experiment are shown in Table 5. The "Types of noise" column in the table lists the noise addition operations described below, and the "Parameters" column lists the noise parameters.

- "Normal" $\cdots$ $s_{\mathrm{add}} \leftarrow s^* + \nu, \nu \sim N(\mu, \sigma^2)$
- "Label" $\cdots$ $s_{\mathrm{add}} \leftarrow \min(\max(s^* + \nu, 0), 1)$, $\nu$ is randomly sampled from $[-1, 0, 1]$ with probability $[p_a, p_b, p_c]$

Table 5: Additive noise types and parameters for each dimension.

| Dimension in state | Types of noise | Parameters |
|---|---|---|
| 1 | Normal | $N(0, 0.01)$ |
| 2 | Normal | $N(0, 0.01)$ |
| 3 | Normal | $N(0, 0.01)$ |
| 4 | Normal | $N(0, 0.01)$ |
| 5 | Normal | $N(0, 0.0001)$ |
| 6 | Normal | $N(0, 0.0001)$ |
| 7 | Normal | $N(0, 0.01)$ |
| 8 | Normal | $N(0, 0.01)$ |
| 9 | Label | $[0.1, 0.8, 0.1]$ |
| 10 | Normal | $N(0, 0.0001)$ |
| 11 | Normal | $N(0, 0.0001)$ |
| 12 | Normal | $N(0, 0.01)$ |
| 13 | Normal | $N(0, 0.01)$ |
| 14 | Label | $[0.1, 0.8, 0.1]$ |
| 15 | Normal | $N(0, 0.01)$ |
| 16 | Normal | $N(0, 0.01)$ |
| 17 | Normal | $N(0, 0.01)$ |
| 18 | Normal | $N(0, 0.01)$ |
| 19 | Normal | $N(0, 0.01)$ |
| 20 | Normal | $N(0, 0.01)$ |
| 21 | Normal | $N(0, 0.01)$ |
| 22 | Normal | $N(0, 0.01)$ |
| 23 | Normal | $N(0, 0.01)$ |
| 24 | Normal | $N(0, 0.01)$ |

## B.2 SENSITIVITY TO NOISE SCALE

To analyze the sensitivity of the learning results to the noise scale, we conducted experiments in which we changed the standard deviation $\sigma$ of the normal distribution $N(0, \sigma^2)$ used to generate the noise. Walker2d in the MuJoCo environment was used for the experiments.

The results are shown in Figure 8. It can be seen that the noise should have a certain level of magnitude and should not be too small or too large. The appropriate noise level is considered to be task-dependent.

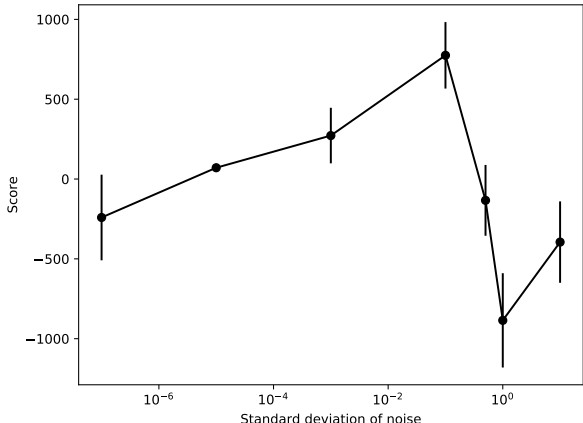

Figure 8: Relationship between noise scale and score in the Walker2d task. The X-axis represents the standard deviation of the normal distribution (logarithmic scale), and the Y-axis represents the RL score (sum of true rewards) based on IRL results. The bars in the graph indicate the standard deviation of the scores.

## B.3 DETAILS OF BEHAVIORAL CLONING

There are various implementation methods for behavioral cloning. We used supervised learning as the simplest behavioral cloning method; specifically, recording the expert's state and action sequences and using the state as input and the action as output.

More specifically, we trained a multilayer perceptron $f(x; \theta)$ for tasks with a discrete action space using the softmax cross-entropy function shown in Eq. (26) as the loss. The number of inputs in the network was $|S|$, the number of outputs was equal to that of action options. The other settings of the network were the same as those of IRL.

$$L(\theta) = -\mathbb{E}_{(s_t, a_t) \sim \mathcal{D}} \left[ a_t \log \text{softmax}\left(f(s_t; \theta)\right) \right] \tag{26}$$

In the evaluation, the argmax policy in Eq. (27) was used:

$$a_t = \text{argmax}_a f(s_t; \theta) \tag{27}$$

For tasks with a continuous action space, the multilayer perceptron was trained by using the squared loss shown in Eq. (28). The number of inputs in the network was $|S|$, and the number of outputs was $|A|$.

$$L(\theta) = \mathbb{E}_{(s_t, a_t) \sim \mathcal{D}} \left[ (a_t - f(s_t; \theta))^2 \right] \tag{28}$$

In the evaluation, the output of the network $f(s_t; \theta)$ was used. Note that the output values were clipped in the defined range of the action in the environment.

## B.4 DETAILS OF RED

To measure the RED scores, we used the author's implementation `https://github.com/RuohanW/RED`. We used the same hyperparameters but changed the number of trajectories to 1000.

