# OpenReview forum: "Deep Inverse Reinforcement Learning via Adversarial One-Class Classification"
_ICLR.cc/2022/Conference — ICLR 2022 Submitted_

### Official Review · Reviewer_uTpc · 2021-11-02

**Correctness:** 3
**Technical Novelty And Significance:** 3
**Empirical Novelty And Significance:** 2
**Recommendation:** 5
**Confidence:** 4

**Main Review:**

### Strengths ###
- The problem that the paper tends to address is challenging and is very relevant to the ICLR community.
- The algorithm proposed seems simple and can be implemented using the off-the-self adversarial one-class algorithms, although tuning noise level may be required as the authors noted.
- The paper is overall clearly written and easy to follow.

### Concerns ###
- The approach that the authors propose is very interesting. However, I am not sure whether the analysis of LogReg-IRL in Section 2.3 carries over to the proposed method. The analysis of LogReg-IRL seems to only apply when learning classifiers that distinguish between trajectories generated from uncontrolled and expert-controlled transition probabilities. It’s not shown in the paper what reward and value functions are learned by learning classifiers that distinguish between expert-controlled and noise-corrupted-export-controlled data. Therefore, theoretical analysis of the proposed method is lacking.
- Do the reward function and transition probability in the experiments satisfy the requirements of LMDP?
- The experiments need more work.
   - Some ablation analysis would help to understand what change contributes to the improvement on the BipedalWalker environment among the addition of generator, weight decay, and changed cross-entropy.
  - LogReg-IRL is affected by baseline trajectories in theory. It would be helpful to compare to related works such as D-REX (Brown et. al. 2019) (an extension of T-REX) that proposes a similar idea of injecting noise to demonstrated trajectories.
  - Due to the randomness in network initialization and trajectory generation, experiments with more seeds are helpful. More tasks would also better demonstrate the performance of the algorithm.
  - I appreciated that the authors are honest about the difficulty in turning the noise levels. Could you give some intuition about this? Moreover, what are the state dimensions in Table 5 in Page 13?
  - It’s exciting to see the visualization of the rewards that SOLO-IRL learned. It will be informative to plot the true reward function too.

 ### Other comments / questions ###
- Eq. (4) (rewriting Eq. (3) using the definition of KL divergence) does not seem correct. And Eq. (5) does not align with the equation in the LogReg-IRL paper.
- Is assuming the states are finite necessary in e.g. Eq. (4) or it’s just for convenience?

### Post-rebuttal comments ###
Thank the authors very much for responding to my questions and adding more experiments in a short amount of time. However, the results do not seem very promising and I am still concerned with the theoretical grounding of the proposed algorithm. Therefore, I would like to keep my score.


**Summary Of The Paper:**

The paper proposes a state-only offline ILR algorithm (learning reward function),  SOLO-IRL, by reducing IRL to adversarial one-class classification. Compared to most existing ILR algorithms, the proposed algorithm is more efficient and requires fewer assumptions. It does not require solving RL problems in the inner loop and does not require ranked expert trajectories or assumptions on trajectories generated by uncontrolled transitions probabilities. The authors show the algorithm learns reasonable reward on two simulated control tasks and significantly outperforms the LogReg-IRL algorithm that it extends.


**Summary Of The Review:**

Because of the concerns discussed in the Main Review tab, I am leaning towards rejecting the paper for now.

---

> ### Author Response · Authors · 2021-11-22
> **Response to Reviewer uTpc**
>
> We thank the reviewer for the many insightful comments. Thanks to the feedback, we have been able to clarify the position of the proposed method. In the following, we respond to each of the comments.
>
> > I am not sure whether the analysis of LogReg-IRL in Section 2.3 carries over to the proposed method. The analysis of LogReg-IRL seems to only apply when learning classifiers that distinguish between trajectories generated from uncontrolled and expert-controlled transition probabilities. It’s not shown in the paper what reward and value functions are learned by learning classifiers that distinguish between expert-controlled and noise-corrupted-export-controlled data. Therefore, theoretical analysis of the proposed method is lacking.
>
> We consider that the proposed method builds on the density ratio estimation framework in LMDP as well as LogReg-IRL; that is, if the appropriate density ratio can be estimated, then the appropriate reward and state-value can be estimated. While the previous study mentioned that discriminator in GANs estimates the density ratio (Uehara et al. 2016), the proposed method employs similar adversarial learning, which is expected to estimate the density ratio according to the distribution of the expert data.
>
> Also, precisely speaking, for expert data, the expert-like data generated by the generator (not the data corrupted by noise) is the target of classification. This avoids the problem of LogReg-IRL, where the appropriate reward cannot be estimated when using a baseline trajectory that does not cover the area where the expert trajectory exists.
>
> > Do the reward function and transition probability in the experiments satisfy the requirements of LMDP?
>
> Since ordinary MDP tasks have been used for evaluation in LogReg-IRL, we consider that the proposed method can also be applied to ordinary tasks.
>
> > Some ablation analysis would help to understand what change contributes to the improvement on the BipedalWalker environment among the addition of generator, weight decay, and changed cross-entropy.
>
> We plan to conduct an ablation study of the proposed method with respect to changing the generator and loss function in the future.
>
> > LogReg-IRL is affected by baseline trajectories in theory. It would be helpful to compare to related works such as D-REX (Brown et. al. 2019) (an extension of T-REX) that proposes a similar idea of injecting noise to demonstrated trajectories.
>
> We have cited D-REX in the related work section to illustrate the differences with the proposed method.
>
> > Due to the randomness in network initialization and trajectory generation, experiments with more seeds are helpful. More tasks would also better demonstrate the performance of the algorithm.
>
> We plan to conduct more experiments with different random seeds as a future work. We have also conducted two new experiments on the MuJoCo task and updated the score table in the paper.
>
> > I appreciated that the authors are honest about the difficulty in turning the noise levels. Could you give some intuition about this? Moreover, what are the state dimensions in Table 5 in Page 13?
>
> In the appendix of our paper, we reported on an experiment to investigate the sensitivity of the proposed method to noise scale. The noise scale is correlated with the range of possible values of the state (in the environment), and the appropriate noise scale will vary depending on the task.
>
> The "dimension in state" in Table 5 (in the appendix) means the value of each sensor in the 24-dimensional state input (observation) of BipedalWalker. For example, the "first dimension" refers to the "hull angle" sensor.
>
> > It’s exciting to see the visualization of the rewards that SOLO-IRL learned. It will be informative to plot the true reward function too.
>
> We plan to investigate how to properly visualize the true rewards of BipedalWalker in the future.
>
> > Eq. (4) (rewriting Eq. (3) using the definition of KL divergence) does not seem correct. And Eq. (5) does not align with the equation in the LogReg-IRL paper.
>
> Thank you for pointing this out. There was an error in Eq. (3), which we have corrected. We consider Eq. (5) to be identical to Eq. (7) in (Todorov 2007).
>
> > Is assuming the states are finite necessary in e.g. Eq. (4) or it’s just for convenience?
>
> Yes, the summation notation is for convenience and can be applied to infinite state space.
>
> (Uehara et al. 2016) Uehara, M., Sato, I., Suzuki, M., Nakayama, K., & Matsuo, Y. (2016). Generative adversarial nets from a density ratio estimation perspective. arXiv preprint arXiv:1610.02920.
>
> (Todorov 2007) Todorov, E. (2007). Linearly-solvable Markov decision problems. In Advances in neural information processing systems (pp. 1369-1376).

---

### Official Review · Reviewer_7Yke · 2021-11-02

**Correctness:** 3
**Technical Novelty And Significance:** 3
**Empirical Novelty And Significance:** 2
**Recommendation:** 3
**Confidence:** 3

**Main Review:**

Overall, this is an interesting paper, but I’m concerned that the experiments don’t compare to prior methods that also learn a reward function purely from expert demonstrations without any unlabeled data:
 - [Random Expert Distillation](https://arxiv.org/abs/1905.06750)
 - [Disagreement-Regularized Imitation Learning](https://openreview.net/forum?id=rkgbYyHtwB)
 - [O-NAIL](https://arxiv.org/abs/2008.03525)

I’m also concerned that the experiments only show that the proposed method outperforms prior work on one task: BipedalWalker. I would consider raising my score if the paper included experiments with different tasks and compared to at least one of the prior methods listed above.

Update
-----
Thank you to the authors for adding the experimental comparisons to BC and RED on the Hopper and Walker2D simulated locomotion tasks. Unfortunately, it seems that the BC and RED baselines substantially outperform the proposed method on those two tasks, so I will keep my original score.


**Summary Of The Paper:**

The paper proposes an adversarial inverse reinforcement learning algorithm that learns purely from expert demonstrations, and does not require any online interaction with the environment or a dataset of unlabeled interactions with the environment. The key idea is to synthesize negative examples (i.e., examples of non-expert behavior) using a denoising autoencoder trained on the positive examples (i.e., expert demonstrations). Experiments on the BipedalWalker simulated locomotion task show that the proposed method learns a reward function such that an RL agent trained to maximize the learned rewards achieves higher true rewards than a prior method.

**Summary Of The Review:**

Lacking comparisons to relevant prior methods and evaluations on diverse tasks

---

> ### Author Response · Authors · 2021-11-22
> **Response to Reviewer 7Yke**
>
> We thank the reviewer for pointing out additional methods and tasks to compare. The suggestions made clear the issues we need to address.
>
> > I’m concerned that the experiments don’t compare to prior methods that also learn a reward function purely from expert demonstrations without any unlabeled data:
>
> > Random Expert Distillation
>
>
> > Disagreement-Regularized Imitation Learning
>
>
> > O-NAIL
>
>
> > I’m also concerned that the experiments only show that the proposed method outperforms prior work on one task: BipedalWalker.
>
> We have updated the experiments section by adding Random Expert Distillation (plus behavioral cloning) as a comparison method and two MuJoCo tasks. We have also listed the three methods in the related work section.

---

### Official Review · Reviewer_phAF · 2021-11-02

**Correctness:** 4
**Technical Novelty And Significance:** 3
**Empirical Novelty And Significance:** 2
**Recommendation:** 6
**Confidence:** 2

**Main Review:**

### Strengths
This work demonstrates an interesting and insightful extension to IRL frameworks. The authors created an efficient IRL method that can be trained in a completely offline manner. By replacing the binary classification with adversarial one-class classifiction from LogReg-IRL, they remove the difficulty of picking a good baseline trajectory that LogReg-IRL requires. Furthermore, this idea of adversarially learned one-class classifier draws on the idea of anomaly detection problem to learn a good discriminator; this can potentially have benefits for a stronger baseline trajectory during training. Indeed, this was demonstrated in the results in the BipedalWalker task, where the random policy could not provide an appropriate baseline trajectory for the expert trajectory to progress to the goal.

### Weaknesses
The main weakness of this work is the sparsity of the experimental section. A baseline that should be compared to is behavioral cloning (BC). The setting of this work is under an offline setting using only expert trajectories which is the same for BC; thus, I am curious if BC performs on-par with SOLO-IRL? The authors do not explicitly mention an advantage of their method over simple BC (although I would imagine the reward function may be an advantage), and thus I would like to see it's performance compared to BC. The authors demonstrate that the reward function is better shaped than LogReg-IRL, but they do not showcase how it can be used. It would be interesting to see if the learned reward can be used to transfer to like tasks?

One additional concern is the sparsity in tasks and baselines in the experimental section. It would be interesting to see more than 1 baseline and tasks other than BipedalWalker/CartPole (since CartPole is quite simple). The authors also mentioned that adjusting the noise in SOLO-IRL is difficult. It would be nice to see an ablations section to understand how sensitive/hard to tune that is, as compared to selecting baseline trajectories?

### Miscellaneous
A more extensive background of the IRL field would be nice. A few citations to include:
 - MaxEnt-IRL, Ziebart et al. 2008
 - Guided Cost Learning, Finn et al. 2016
 - Generative adversarial imitation learning, Ho et al. 2016
 - Behavioral Cloning from Observation, Torabi et al. 2018

**Summary Of The Paper:**

This work introduces a new IRL framework, SOLO-IRL, that learns a reward function using only expert trajectories. This has the benefit of being trained in an offline manner, which speeds up the training process. SOLO-IRL builds on top of the work of Uchibe, 2018 and exploits the fact that a discriminator can replace the binary classification used in LogReg-IRL. They improve on LogReg-IRL by overcoming the difficulty of selecting an appropriate baseline trajectories by proposing to use adversarial one-class classification (Sabokrou et al., 2018). The authors empirically demonstrate their results on the CartPole and BipedalWalker tasks, and show superior performance over LogReg-IRL.

**Summary Of The Review:**

The authors introduce an interesting, new framework for IRL that can learn a reward function in an offline manner. Furthermore, they draw insights from anomaly detection problem and improve upon the LogReg-IRL. They demonstrate, empirically, that their method can learn a better reward function and achieve higher performance on two OpenAI Gym tasks. However, the experimental section for this work is very sparse; if the authors can address the concerns listed above, in particular adding BC as a baseline, I would be willing to increase my rating.

---

> ### Author Response · Authors · 2021-11-22
> **Response to Reviewer phAF**
>
> We thank the reviewer for the very helpful feedback. We have gained a lot of new insights to improve the paper. We respond to each comment below.
>
> > A baseline that should be compared to is behavioral cloning (BC). The setting of this work is under an offline setting using only expert trajectories which is the same for BC; thus, I am curious if BC performs on-par with SOLO-IRL? The authors do not explicitly mention an advantage of their method over simple BC (although I would imagine the reward function may be an advantage), and thus I would like to see it's performance compared to BC.
>
> As you pointed out, behavioral cloning (BC) is an appropriate offline imitation learning method as a baseline for SOLO-IRL. We have therefore implemented and experimented with a simple BC and updated the score table in the paper (implementation details are in the appendix).
>
> As a result, BC scored better than SOLO-IRL in the task where the environment did not change. On the other hand, the proposed method scored better in the BipedalWalker task, where the environment (such as the ground shape) changed each time slightly. Based on these results, we considered that the proposed method is robust to changes in the environment (in other words, the environment in which covariate shifts occur).
>
> > The authors demonstrate that the reward function is better shaped than LogReg-IRL, but they do not showcase how it can be used. It would be interesting to see if the learned reward can be used to transfer to like tasks?
>
> Reward transferability is an important issue for IRL. We would like to include in our future work the search for and validation of similar tasks (with the same number of dimensions of state input and action output) that can be easily transferred.
>
> > One additional concern is the sparsity in tasks and baselines in the experimental section. It would be interesting to see more than 1 baseline and tasks other than BipedalWalker/CartPole (since CartPole is quite simple). The authors also mentioned that adjusting the noise in SOLO-IRL is difficult. It would be nice to see an ablations section to understand how sensitive/hard to tune that is, as compared to selecting baseline trajectories?
>
> We conducted additional experiments and added the scores on two MuJoCo tasks. We also added BC and Random Expert Distillation (Wang et al. 2019) as a comparison method. In addition, we experimented with the sensitivity of the proposed method to the noise scale in the Walker2d environment. The results are described in the appendix section. It turned out to be more sensitive to the noise scale than we had assumed.
>
> > A more extensive background of the IRL field would be nice. A few citations to include:
>
> We have included them in the Related Work section.
>
> (Wang et al. 2019) Wang, R., Ciliberto, C., Amadori, P. V., & Demiris, Y. (2019, May). Random expert distillation: Imitation learning via expert policy support estimation. In International Conference on Machine Learning (pp. 6536-6544). PMLR.

---

### Decision · Program_Chairs · 2022-01-20

**Decision:**

Reject

**Comment:**

This paper studies the problem of inverse reinforcement learning by relying on only demonstrations and no interaction (like imitation learning). The reviewers liked the premise but had major concerns with evaluation and baselines. The paper initially received reviews tending to reject. One of the questions was about missing behavior cloning baseline which the authors added in rebuttal. But the BC baseline seems to be really competitive (in fact, better in 3 out of 4 envs) as compared to the proposed approach. In conclusion, all reviewers still believed that their concerns regarding insufficient evidence for justifying approach and missing comparisons to other prior work still stand. AC agrees with the reviewers' consensus that the paper is not yet ready for acceptance.